

**Projected Climate Change Impacts on Future Streamflow of the Yarlung**
**Tsangpo-Brahmaputra River**
Ran Xu[1], Hongchang Hu[1], Fuqiang Tian[1*], Chao Li[2], Mohd Yawar Ali Khan[1]
*1. Department of Hydraulic Engineering, Tsinghua University, Beijing 100084, China*
*2. Pacific Climate Impact Consortium, University of Victoria, Victoria, British Columbia, V8W*
*2Y2, Canada*
**Corresponding Author:** Fuqiang Tian (tianfq@tsinghua.edu.cn)



## Abstract

The Yarlung Tsangpo-Brahmaputra River (YBR) originating from the Tibetan Plateau (TP), is an important water source for many domestic and agricultural practices in countries including China, India, Bhutan and Bangladesh. To date, only a few studies have investigated the impacts of climate change on water resources in this river basin with dispersed results. In this study, we provide a comprehensive and updated assessment of the impacts of climate change on YBR streamflow by integrating a physically based hydrological model, regional climate integrations from CORDEX (Coordinated Regional Climate Downscaling Experiment), different bias correction methods, and Bayesian model averaging method. We find that (i) bias correction is able to reduce systematic biases in regional climate integrations and thus benefits hydrological simulations over YBR Basin; (ii) Bayesian model averaging, which optimally combines individual hydrological simulations obtained from different bias correction methods, tends to provide hydrological time series superior over individual ones. We show that by the year 2035, the annual mean streamflow is projected to change respectively by 6.8%, -0.4%, and -4.1% under RCP4.5 relative to the historical period (1980-2001) at the Bahadurabad in Bangladesh, the upper Brahmaputra outlet, and Nuxia in China. Under RCP8.5, these percentage changes will substantially increase to 12.9%, 13.1%, and 19.9%. Therefore, the change rate of streamflow shows strong spatial variability along the YBR from downstream to upstream. The increasing rate of streamflow shows an augmented trend from downstream to upstream under RCP8.5 compared to an attenuated pattern under RCP4.5.

**Keywords:** Climate Change Impacts, Yarlung Tsangpo-Brahmaputra River, Streamflow, Regional Climate Integrations, Bias Correction, Bayesian Model Averaging





## 1. Introduction

Water is a standout necessity amongst the most basic factors in human sustenance (Barnett et al.,
2005). Global climate change has been found to intensify the global hydrological cycle, likely
creating predominant impacts on regional water resources (Arnell, 1999; Gain et al., 2011).
Evaluation of the potential impacts of anthropogenic climate change on regional and local water
resources relies largely on climate model projections (Li et al., 2014). The spatial resolution of
typical global climate models (GCMs) (100–300 km) is insufficient to simulate regional events
that are needed to capture different climate and weather phenomena at regional to local scales
(e.g., the watershed scale) (Olsson et al., 2015). Climate simulations from GCMs can be
dynamically downscaled with regional climate models (RCMs) to scales of 25–50 km. Despite
that dynamical downscaling is computationally very demanding and that its accuracy depends to
a large extend on that of its parent GCM, dynamical downscaling can provide more detailed
information on finer temporal and spatial scales than GCMs (Hewitson and Crane, 1996). Such
information is valuable for impact projections at regional to local scales that are more relevant to
water resources management.
On the other hand, although the increased horizontal resolution can improve the simulation of
regional and local climate features, RCMs still produce biases in the time series of climatic
variables (Christensen et al., 2008; Rauscher et al., 2010). Bias correction is typically applied to
the output of climate models. Most bias correction methods correct variables separately, with
interactions among variables typically not considered (Christensen et al., 2008; Hessami et al.,
2008; Ines and Hansen, 2006; Johnson and Sharma, 2012; Li et al., 2010; Piani et al., 2009; Piani
et al., 2010). Separate-variable bias correction methods, for example, may result in physically



unrealistic corrections (Thrasher et al., 2012) and do not correct errors in multivariate
relationships (Dosio and Paruolo, 2011). Correspondingly, Li et al. (2014) introduced a joint bias
correction (JBC) method and applied it to precipitation (P) and temperature (T) fields from the
fifth phase of the Climate Model Intercomparison Project (CMIP5) model ensemble.
The Yarlung Tsangpo-Brahmaputra River (YBR) is an important river system originating from
the Tibetan Plateau (TP), characterized by a dynamic fluvial regime with exceptional
physiographic setting spread along the eastern Himalayan region (Goswami, 1985). Critical
hydrological processes like snow and glacial melt are more important in this area compared to
others. Hydrological processes of the YBR Basin are highly sensitive to changes in temperature
and precipitation, which subsequently affect the melting characteristics of snowy and glaciered
areas and thus affect streamflow. The YBR Basin is also one of the most under-investigated and
underdeveloped basins around the world, with only few studies examined the impacts of climate
change on the hydrology and water resources of this basin (Immerzeel et al., 2010; Lutz et al.,
2014; Masood et al., 2015). Immerzeel et al. (2010) developed a snowmelt-runoff model in the
upper YBR Basin using native output (without bias correction) from 5 GCMs under the A1B
scenarios for 2046-2065 and found that its streamflow would decrease by 19.6% relative to
2000-2007. Subsequently, Lutz et al. (2014) implemented the SPHY (Spatial Processes in
Hydrology) hydrological model in the upper YBR Basin using native simulations from 4 GCMs
under RCP4.5 and RCP8.5 emissions scenarios for 2041-2050 and showed that the streamflow
would increase by 4.5% and 5.2% relative to 1998-2007 under the examined two emissions
scenarios. Masood et al. (2015) applied the H08 Hydrological model to the YBR Basin using
bias corrected projections of 5 GCMs for near future (2015-2039) and far future (2075-2099)



periods and found that relative to the period 1980-2001, the streamflow would increase by 6.7%
and 16.2% for near and far future under RCP8.5, respectively.
Several factors could contribute to the discrepancy between these projections, such as the lack of
high quality streamflow observations for hydrological model calibration, the choice of bias
correction methods, simulations from global climate models, and future emissions scenarios, and
a combination thereof. On the other hand, all the existing studies in the YBR basin rely on
GCMs, which, as was discussed, cannot capture fine-scale climate and weather details that are
required for a reliable regional impacts assessment. In the present study, we attempt to fill this
gap by taking advantage of the recently compiled multi-model and multi-member high-resolution
regional climate integrations from CORDEX (Coordinated Regional Climate Downscaling
Experiment). We use different bias correction methods to alleviate the inherent biases in these
regional climate integrations, and use a Bayesian model averaging technique to best combine
different streamflow simulations obtained with different bias-corrected meteorological forcing
data (e.g., precipitation and temperature). We synthesize our results and those in the existing
studies with a hope to obtain a more comprehensive picture of changes in water resources in the
YBR Basin in response to global climate warming.
We structured the paper into the following sections. Section 1 formulates the objectives of this
study. Section 2 briefly introduces the YBR Basin, followed by the used materials and methods.
Our results and those in existing studies are compared in Section 3. Main conclusions along with
a brief discussion of the future scope of this study are presented in Section 4.

## 2. Materials and methodology

### 2.1 The Yarlung Tsangpo-Brahmaputra River Basin



Tibetan Plateau (TP) is often referred as Asia's water tower, bordered by India and Pakistan in
the west side and Bhutan and Nepal on the southern side, with a mean elevation of about 4000 m
above sea level (Tong et al., 2014). The YBR is one of the largest rivers originating from the TP
in Southwest China at an elevation of about 3100 m above sea level (Goswami, 1985; Xu et al.,
2017). The total length of the river is about 2900 km (Masood et al., 2015), with a drainage area
of the basin estimated to be around 530,000 km$^2$. The YBR travels through China, Bhutan, and
India before emptying into the Bay of Bengal in Bangladesh (Figure 1). The mean annual
discharge is approximately 20,000m$^3$/s (Immerzeel, 2008). The climate of the basin is
monsoon-driven with an obvious wet season from June to September, which accounts for 60–70%
of the annual rainfall.
**2.2 Data**
**2.2.1 Forcing data sets**
Due to the lack of adequate in-situ meteorological observations, the WATCH forcing data (WFD)
(Weedon et al., 2014) were used as a reference for bias correction and hydrological model
calibration (Table 1). This dataset provided a good representation of real meteorological events
and climate trends (Weedon et al., 2011). In this study, we used daily rainfall, temperature and
potential evapotranspiration (*PET*) data from 1980 to 2001.
The sources of other required non-meteorological variables for implementing the hydrological
model are as follows. The 90-m resolution digital elevation model data were acquired from the
Shuttle Radar Topography Mission (SRTM) (http://srtm.csi.cgiar.org/). The Leaf Area Index
(LAI) and snow cover data from 2000 to 2016 were downloaded from the National Aeronautics
and Space Administration (NASA) (https://reverb.echo.nasa.gov/reverb/). For the periods during
which LAI and snow data did not cover, average values of LAI and snow were used as model



input. The biweekly normalized difference vegetation index (NDVI) data from 1982 to 2000
with a spatial resolution of 8 km were obtained from the Global Inventory Modeling and
Mapping    Studies-Advanced    Very    High    Resolution    Radiometer    (GIMMS-AVHRR)
(http://www.glcf.umd.edu/data/gimms/). The soil hydraulic parameters were derived from the
soil classification data which were extracted from the global digital soil map with a spatial
resolution of 10 km (http://www.fao.org/geonetwork/).
**2.2.2 Hydrological data**
The streamflow observations during 1980-2001 for hydrological model calibration were obtained
at two hydrological stations, i.e., the Nuxia station located in upstream China (Gao et al. (2008))
and the Bahadurabad station located in downstream Bangladesh; see Figure 1 for their
geographical locations.
**2.2.3 RCM data**
The simulations of daily precipitation and temperature during the historical period of 1980-2001
and the projections under two examined emissions scenarios (RCP4.5 and RCP8.5) during the
future period of 2020-2035 from the CORDEX experiment for the East Asia domain (which
covers the whole YBR Basin) were downloaded from http://www.cordex.org/. The CORDEX
program, which was coordinated by the World Climate Research Program, provides a unique
opportunity for generating high-resolution regional climate projections and for assessing the
impacts of future climate change at regional scales (Piani et al., 2009). As shown in Table 1,
climate data from 5 CORDEX models were chosen. These models include HadGEM3-RA
(denoted by RCM1), RegCM4 (RCM2), SNU-MM5 (RCM3), SNU-WRF (RCM4) and
YSU-RSM (RCM5). To keep consistent with the WATCH forcing data, climate model
integrations were interpolated to the grid of the WFD using the bilinear interpolation method.



The adopted hydrological model, as will be introduced later, also requires *PET* as a forcing
variable. We used the method proposed by Leander and Buishand (2007) and S. C. van Pelt
(2009) to calculate *PET* with daily temperature *T* as follows:

$$PET = [1 + \alpha_0 (T - \overline{T_0})] \, \overline{PET_0} \qquad (1)$$

where $\overline{T_0}$ is the observed mean temperature (°C) and $\overline{PET_0}$ is the observed mean $PET_0$
(mm/day) during the historical period. Daily $PET_0$ data were acquired directly from the WFD
dataset and were used to compute $\overline{PET_0}$. The proportionality constant $\alpha_0$ was determined for
each calendar month by regressing the observed *PET* at each grid cell onto the observed daily
temperature.
**2.3 Methodology**
**2.3.1 Hydrological model: THREW**
We adopted the Tsinghua Representative Elementary Watershed (THREW) model (Tian, 2006;
Tian et al., 2006) to simulate streamflow of the YBR Basin. The model consists of a set of
balance equations for mass, momentum, energy and entropy, including associated constitutive
relationships for various exchange fluxes, at the scale of a well-defined spatial domain. Details of
the model can be found in Tian et al. (2006). The THREW model has been successfully applied
to quite a few watersheds across China and United States (Li et al., 2012; Mou et al., 2008; Sun
et al., 2014; Tian et al., 2006; Tian et al., 2012; Xu et al., 2015; Yang et al., 2014). For the
simulation of snow and glacier melting processes which is important for the YBR Basin, we
modify the original THREW model by incorporating the temperature-index method introduced
in Hock (2003). The index-temperature method has been shown to exhibit an overall good
performance in mountain areas in China (He et al., 2015).



### 2.3.2 Bias correction methods

Quantile mapping (QM) with reference observations has been routinely applied to correct biases in RCM simulations (Maraun, 2013). Using WFD as reference observations and following the principle of QM, first we estimated cumulative distribution functions (CDFs) for the observed and native RCM-simulated time series of daily precipitation or temperature during the historical/calibration period (which is 1980-2001 in this study); then for a given RCM-simulated data value from an application period (which may be historical 1980-2001 period or future 2020-2035 period), we evaluated the CDF of the native RCM simulations at the given data value, followed by evaluation of the inverse of the CDF of the observations at the thus obtained CDF value; the resulting value is the bias-corrected simulation (see Figure 2 for an schematic illustration of this procedure).

Independent bias correction for multiple meteorological variables can produce non-physical corrections. To alleviate the deficits of independent bias correction, Li et al. (2014) introduced a joint bias correction (JBC) method, which takes the interactions between precipitation and temperature into account. This approach is based on a general bivariate distribution of P-T and essentially can be seen as a bivariate extension of the commonly used univariate QM method. Depending on the sequence of correction, there are two versions of JBC including JBCp, which corrects precipitation first and then temperature, and JBCt, which corrects temperature first and then precipitation. For more details of the QM and JBC methods, readers can refer to Wlicke et al. (2013) and Li et al. (2014), respectively.

### 2.3.3 Bayesian model averaging method

Bayesian model averaging (BMA) is a statistical technique designed to infer a prediction by weighted averaging predictions from different models/simulations. We refer readers to Dong et



al. (2013), which have presented a nice description of the basic principle of this method and the
Expectation-Maximization (EM) algorithm for optimally searching the BMA weights. Several
studies have applied BMA to RCMs or GCMs simulations to assess climate change impacts on
hydrology with meaningful results (Bhat et al., 2011; Duan et al., 2007; Wang and Robertson,
2011; Yang et al., 2011).

## 3. Results and discussion


### 3.1 Bias correction of meteorological variables during the historical period


We applied the three bias correction methods (i.e., QM, JBCp and JBCt) to the CORDEX
simulations of daily precipitation and temperature. We found that without bias correction, the
native RCM1 and RCM2 simulations (see Table 1 for the full names of different RCMs)
overestimate precipitation for all months during the 1980-2001 baseline period (Figure 3a-3b),
while native simulations by the other models tend to overestimate precipitation of the dry-season
(November to May of next year) and underestimate precipitation of other months. After bias
correction, the above mentioned overestimation and underestimation reduces considerably. For
temperature, we found that all the examined climate models consistently exhibit cold biases
across all the months, and that such biases are largely eliminated after applying bias correction
(Figure 4). In general, the three bias correction methods exhibit similar skills in reducing
temperature biases (Table 2), with JBCt and QM showing somewhat better performance than
JBCp. As expected, *PET* calculated from bias-corrected temperature simulations was quite close
to WFD observations.
In summary, we found that almost all the bias correction methods can substantially reduce biases
for all the three variables across the months, though with sizeable variations between bias





correction methods and across variables and seasons, consistent with existing studies on the
comparison of different bias correction methods (Maraun, 2013; Prasanna, 2016).
**3.2 Hydrological model setup and simulation**
To setup the THREW model, the whole basin was discretized into 237 representative elementary
watersheds (REWs). There are in total 16 parameters involved in THREW, as listed in Table 3.
The first 6 parameters were determined for each REW a prior from the data described in the
section 'Materials and methodology'. The remaining parameters were subjected to calibration
and assumed to be uniform across the 237 REWs. Automatic calibration was implemented by the
ε-NSGAII optimization algorithm developed by Reed et al. (2003). We chose the commonly
used Nash Sutcliffe efficiency coefficient (NSE) (Nash and Sutcliffe, 1970) as the single
objective function for model calibration.
We divided the whole period 1980-2001 into two sub-periods, which were used respectively for
model calibration (1980-1990) and validation (1991-2001). Simulated daily streamflow time
series at Bahadurabad were compared against the corresponding observations to compute the
NSE objective function. To warm up the model, we dropped the first year of the calibration
period (i.e., 1980). Observed and simulated daily streamflow of remaining years were used to
compute NSE as follows:

$$\text{NSE} = 1 - \frac{\sum_{n=1}^{N} (Q_o^n - Q_s^n)^2}{\sum_{n=1}^{N} (Q_o^n - \overline{Q_o})^2} \tag{2}$$

where $N$ denotes the total number of days in the calibration period (which is 1981-1990 as one
year is dropped for model warming up); $Q_o^n$ and $Q_s^n$ represent respectively the observed and
simulated streamflow of day $n$; and $\overline{Q_o}$ is the average of observed streamflow during that period.
NSE is automatically optimized by the ε-NSGAII optimization algorithm. With the calibrated



model, NSE for the 1991-2001 validation period can be likewise computed so as to assess the
calibrated model performance in simulating streamflow that is not seen in the calibration period.
Figure 5 shows the observed (black line) and simulated (red line) discharges at Bahadurabad at
(a) daily, (b) monthly, and (c-d) seasonal time scales for both the calibration and validation
periods. It can be seen that the THREW model performs well in the YBR Basin at all time scales.
During the calibration period the daily and monthly NSE values are 0.84 and 0.92, respectively,
and during the validation period the daily and monthly NSE values are 0.78 and 0.84,
respectively. We also compared the observed and simulated monthly discharges at the Nuxia
station, which is not involved in model calibration. The monthly NSE values of calibration and
validation periods were 0.66 and 0.73, respectively. In summary, these results suggest that the
THREW model does a good job in simulating the hydrological processes in the YBR Basin
during this historical period. We assume that the calibrated THREW model is applicable to the
future period. This assumption is necessary in this study and has been widely adopted in previous
climate impacts studies.
Figure 6 compares the seasonal streamflow simulated by the THREW model with observed
streamflow data at Bahadurabad. It is observed that the streamflow generated by native RCM
simulations tends to either over- or underestimate the observations, and that all the adopted bias
correction methods can alleviate, to varying degrees, these biases. We found that in general bias
correction is more effective in improving the simulation of dry season streamflow (from
November to April in the next year) than that of wet season (May to October). Table 4 shows the
annual mean observed streamflow at Bahadurabad as well as the simulated streamflow with the
WFD data and with the native and bias-corrected RCM integrations. We can see that at annual
scale, streamflow simulated with native RCMs is on average higher (e.g., RCM1, RCM2) or



lower (e.g., RCM3, RCM4 and RCM5) than the observations; while streamflow simulated with
bias-corrected RCMs is much more consistent with the observations.
Table 5 presents the NSE values for the daily and monthly streamflow over the calibration and
validation periods simulated by the THREW model with the WFD data and with native and
bias-corrected RCM simulations at Bahadurabad. We found that QM and JBCp can improve
NSE for almost all the RCMs except RCM5, while JBCt can improve NSE for three of the five
climate models (RCM1, RCM3, and RCM4). We also found that none of the 3 bias correction
methods is compelling better than others, suggesting the necessity of combining different
streamflow simulations generated with different bias-corrected climate simulations. Moreover, it
is seen that most of the NSEs values are higher than 0.55 with a few exceptions, indicating
reasonably well simulations of daily and monthly streamflow for both calibration and validation
periods on average across the entire basin, and thus enhancing our confidence in applying the
calibrated THREW model and the bias-corrected CORDEX simulations to projecting future
hydrological conditions in the YBR Basin.
Given the fact that none of the bias correction methods and none of the RCM models are
compellingly superior over others, as we have found, we therefore integrate streamflow
simulations generated by different bias-corrected climate simulations from different climate
models with different bias correction methods in terms of BMA. Our attempt is to take
advantages of individual streamflow simulations. Daily streamflow simulations and observations
during the THREW model calibration period (1981-1990) were used to calibrate the BMA
weights, and those during the validation period are used to evaluate the calibrated BMA weights.
In addition to NSE, two other indices were used to measure the closeness between observations



and simulations. These indices are relative error (RE) and root mean squared error (RMSE), both
evaluated at daily scale, as defined in the following:

$$\text{RE} = 1 - \frac{\sum_{n=1}^{N} Q_s^n}{\sum_{n=1}^{N} Q_o^n} \tag{3}$$

$$\text{RMSE} = \sqrt{\frac{\sum_{n=1}^{N}(Q_o^n - Q_s^n)^2}{N}} \tag{4}$$

where $N$ denotes the total number of days during the considered period; $Q_o^n$ and $Q_s^n$ represent
respectively the observed and simulated streamflow of time $n$. As seen from Table 6, based on
the above indices, after applying BMA we obtain considerably better results than almost all those
generated by different bias-corrected climate simulations from different climate models with
different bias correction methods. Figure 7 shows the mean prediction (red line) and 90%
uncertainty interval of BMA during the historical period at Bahadurabad. The uncertainty
interval of BMA can cover almost all observations, which further indicated the sound
performance of BMA.
**3.3 Projections of future meteorological variables**
Figures 8-9 show changes in seasonal precipitation and temperature during the near future period
2020-2035 relative to the historical 1980-2001 period based on bias-corrected RCM simulations
under RCP4.5 and RCP8.5 emissions scenarios. It is found that precipitation in wet seasons will
increase under both emissions scenarios and in all bias-corrected RCM simulations with one
exception of RCM3 under RCP4.5. In contrast, precipitation in dry seasons is projected to
consistently decrease in all the studied RCM models. Therefore, the general pattern of "wet
getting wetter, dry getting drier" (Chou et al., 2013) associate with climate change exists in YBR
as well. Also, as expected, precipitation under RCP8.5 is on average higher than that under





RCP4.5, especially for RCM3 and RCM4 in the wet season. We also found obvious variations in
the projected changes among climate models and bias correction methods. This suggests the
importance of exploring multi-models and multi-methods to obtain a more comprehensive
picture about the uncertainty of the impacts of climate change on local hydrology. Using BMA
weight coefficient calculated in Section 3.2, weighted precipitation in historical period, RCP4.5
and RCP8.5 is 1425.3, 1529.8 and 1608.0 mm per year, respectively.
We found that temperature is projected to increase by all RCM simulations in both dry seasons
and wet seasons (Figure 9). It is surprising to see that there is no significant difference in
temperature between RCP8.5 and RCP4.5 scenarios except for RCM3 and RCM4. In fact, this is
not inconsistent with the IPCC AR5 (2013), which shows that the projected future global mean
temperature does not significantly diverge under different RCP scenarios until 2030 (our future
period is 2020-2035). Similar to precipitation, there are obvious variations in the projected
changes among different climate models and different bias correction methods. Using BMA
weight coefficient calculated in Section 3.2, weighted temperature in historical period, RCP4.5
and RCP8.5 is 8.7, 9.8 and 10.0℃, respectively.

**3.4 Projections of future streamflow and comparison with previous studies**

Figure 10 shows the mean prediction and 90% uncertainty interval of streamflow simulated by
BMA method during (a) RCP4.5, (b) RCP8.5 scenarios at Bahadurabad. Uncertainty interval of
RCP4.5 is similar with that of RCP8.5. All the following discussions in this subsection is based
on BMA weighted streamflow.
For the sake of comparison between Immerzeel et al. (2010), Lutz et al. (2014), Masood et al.
(2015) and our results, we also examined an upstream outlet location (the red dot in Figure 1),



which was studied in the referred studies. To be noted, the observed streamflow data at this
upstream outlet are unavailable.
Table 7 shows a summary of the referred existing studies about climate impact on future
streamflow in the YBR Basin. Immerzeel et al. (2010) developed Snowmelt Streamflow Model
for the upper YBR Basin using five GCMs in the A1B scenarios defined in IPCC AR4 during
2046-2065 without applying any bias correction methods or BMA method and the streamflow
will decrease by 19.6% when compared to the observed period (2000-2007). The SPHY model
developed by Lutz et al. (2014) for the upper YBR Basin using four GCMs in the RCP4.5 and
RCP8.5 scenarios during 2041-2050 and without applying any bias correction methods or BMA
method. The streamflow will increase by 4.5% and 5.2% in the RCP4.5 and RCP8.5 scenarios,
respectively when compared with the observed period (1998-2007). Masood et al. (2015) applied
H08 Hydrological model the YBR Basin using five GCMs during the near future (2015-2039)
and far future (2075-2099) and also applied bias correction method. The streamflow increased by
6.7% and 16.2% in the near future and far future, respectively, when compared with the observed
data (1980-2001).
The comparisons among the streamflow projection of YBR during different periods in different
studies are shown in Figure 11. In our study, the projected streamflow is 1466 mm/a during
2020-2035 under RCP8.5 at Bahadurabad, which is substantially higher than the findings of
Masood et al. (2015) at the same location, which is 1244 mm per year during 2015-2039 under
RCP8.5. The projected streamflow is 692 mm per year during 2020-2035 under RCP8.5 at the
upper YBR outlet. This result is quite close to the findings of Lutz et al. (2014), which is 727
mm per year during 2041-2050 under RCP8.5. To be noted, our study adopted RCMs
integrations, BMA method by incorporating different bias correction methods, and a physically





based hydrological model accounting for snow and glacier melting processes, which could
explain the differences from the existing studies.
Table 8 shows the relative changes of projected runoff and its driving factors under different
emission scenarios compared to the historical period at different locations of the YBR. At the
basin-wide scale represented by Bahadurabad station, future streamflow shows an evidently
increasing trend under both RCP4.5 and RCP8.5 scenarios. The increasing rate under RCP8.5
(12.9%) is not-surprisingly higher than RCP4.5 (6.8%). Also, the trends of streamflow exhibit
strong spatial variability along the YBR. Under RCP4.5, upstream locations are more likely to
experience an increasing trend at a much less rate. For example, the change rate of streamflow is
projected to decrease at 0.4% and 4.1% at the YBR outlet and Nuxia, respectively. Under
RCP8.5, however, upstream locations would more likely witness an augmented increasing rate of
streamflow change, e.g., 13.1% and 19.9% at the YBR outlet and Nuxia, respectively.

## 353    4. Conclusions

In this study, we conducted a comprehensive evaluation of future streamflow in the YBR Basin.
We adopted RCMs integrations, BMA method by incorporating different bias correction
methods, and a physically based hydrological model accounting for snow and glacier melting
processes to implement the evaluation. The major findings are summarized as follows.
(1) The three bias correction methods implemented in this study can all substantially reduce

biases in the three variables (precipitation, temperature and potential evapotranspiration).

Specifically for precipitation, when native RCMs show overestimations, all bias correction

methods perform reasonably well. While, none of them can provide satisfying corrections

when native RCMs exhibit strong underestimations. This finding is consistent with existing





studies (Maraun, 2013; Prasanna, 2016) and requires further in-deep studies. For
temperature and potential evapotranspiration, all of the three bias correction methods
performed well, especially QM and JBCt.
(2) The basin-wide discharge is projected to increase substantially during the future period
(2020-2035) under the two examined emissions scenarios of RCP4.5 and RCP8.5. The
projected annual mean streamflow at Bahadurabad is 1386.7 mm per year under RCP4.5
with an increasing rate of 6.9%, and the number becomes higher as 1466.4 mm per year
under RCP8.5 with an increasing rate of 12.9%. Increasing mean annual streamflow
indicates more flood events that would occur in this already vulnerable region, which calls
for more close collaborations among upstream and downstream riparian countries.
(3) Projected streamflow exhibits different spatial patterns under different scenarios in the YBR
basin. Under RCP4.5, the annual mean streamflow is projected to change by 6.8%, -0.4%,
and -4.1% in the future period (2020-2035) compared to the historical period (1980-2001) at
three locations from downstream to upstream along the YBR, i.e., Bahadurabad, the upper
YBR outlet, and Nuxia. Therefore, the increasing rate of streamflow exhibits an attenuated
trend from downstream to upstream. Under RCP8.5, however, the increasing rate of
streamflow (12.9%, 13.1%, and 19.9% at the three locations) exhibits an augmented trend
from downstream to upstream. The different trends are likely associated with varying spatial
patterns of projected future precipitation, but more detailed investigations are needed.



## Acknowledgements

This study was financially supported by the National Science Foundation of China (91647205), the Ministry of Science and Technology of P.R. China (2016YFA0601603, 2016YFC0402701), and the foundation of State Key Laboratory of Hydroscience and Engineering of Tsinghua University (2016-KY-03).




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



## List of Tables






Table 1. Description of the WATCH forcing data and 5 RCM datasets.

| Type | Dataset | Spatial resolution | Temporal resolution | Period | Description |
|------|---------|--------------------|--------------------|--------|-------------|
| Observation data | WATCH Forcing Data (WFD) | 0.5˚ | Daily | 1980-2001 | Rainfall, air temperature, potential evapotranspiration |
| RCM data | HadGEM3-RA (RCM1)<br><br>RegCM (RCM2)<br><br>SNU-MM5 (RCM3)<br><br>SNU-WRF (RCM4)<br><br>YSU-RSM (RCM5) | 0.44˚ | Daily | 1980-2001<br><br>2020-2035 (RCP4.5, RCP8.5) | Rainfall, air temperature, surface pressure, specific humidity |




Table 2. Annual mean values of basin-wide precipitation (ppt), temperature (tmp) and potential
evapotranspiration (pet) calculated from WFD and native/corrected RCMs datasets.

|  |  | native | JBCp | JBCt | QM |
|---|---|---|---|---|---|
| ppt | WFD | 1310 |  |  |  |
| mm /yr | RCM1 | 2025 | 1296 | 1283 | 1296 |
|  | RCM2 | 1834 | 1312 | 1299 | 1312 |
|  | RCM3 | 1101 | 1584 | 1726 | 1584 |
|  | RCM4 | 1242 | 1523 | 1617 | 1523 |
|  | RCM5 | 1381 | 1325 | 1338 | 1325 |
| tmp | WFD | 8.77 |  |  |  |
| ℃ | RCM1 | 5.80 | 8.85 | 8.77 | 8.77 |
|  | RCM2 | 4.48 | 8.69 | 8.77 | 8.77 |
|  | RCM3 | 4.99 | 8.23 | 8.77 | 8.77 |
|  | RCM4 | 3.77 | 8.57 | 8.77 | 8.77 |
|  | RCM5 | 0.36 | 8.38 | 8.77 | 8.77 |
| pet | WFD | 532 |  |  |  |
| mm /yr | RCM1 | 448 | 525 | 542 | 542 |
|  | RCM2 | 430 | 528 | 542 | 542 |
|  | RCM3 | 474 | 526 | 553 | 553 |
|  | RCM4 | 479 | 540 | 543 | 543 |
|  | RCM5 | 478 | 513 | 532 | 532 |






Table 3. Principal parameters of THREW model.

| Symbol | Unit | Physical meaning | Range | Calibrated value |
|---|---|---|---|---|
| $K_s^u$ | m/s | Saturated hydraulic conductivity for u-zone which is different for each REW. The value showing here is the averaged value over the whole catchment | - | 6.25e-6 |
| $K_s^s$ | m/s | Similar to $K_s^u$, saturated hydraulic conductivity for s-zone | - | 6.25e-6 |
| $\varepsilon^u$ | - | Soil porosity value of u-zone which is different for each REW. The value showing here is averaged over the whole catchment | - | 0.47 |
| $\varepsilon^s$ | - | Similar to $\varepsilon^u$, soil porosity of s-zone | - | 0.28 |
| $\psi_a$ | m | Air entry value which is different for each REW. The value showing here is averaged over the whole catchment | - | 0.25 |
| $\mu$ | - | Soil pore size distribution index in $\overline{f_e} = \alpha^{EFL} \dfrac{\overline{K_s^u}}{(1-s^u)y^u} \dfrac{(S^u)^{2+d}\varepsilon^u|\psi_a|}{\mu}$, where $\overline{f_e}$ is the exfiltration capacity from u-zone, $s^u$ is the saturation degree of u-zone, $y^u$ is the soil depth of u-zone, $d$ is the diffusion index $(d = 1 + 1/\mu)$. The value showing here is the averaged value over the whole catchment | - | 0.20 |
| $n^t$ | - | Manning roughness coefficient for hillslope | 0.005-1 | 0.03 |
| $n^r$ | - | Similar to $n^r$, Manning roughness coefficient for channel | 0.005-1 | 0.006 |
| $B$ | - | Shape coefficient to calculate the saturation excess streamflow area | 0.1-1 | 0.5 |
| $KKA$ | - | Coefficient to calculate subsurface flow in $R_g = KKD \cdot S \cdot K_s^S \cdot (y^S/Z)^{KKA}$, When $S$ is the topographic slope, $y^s$ is the depth of s-zone, $Z$ is the total soil depth | 1-30 | 5.0 |
| $KKD$ | - | See describe for $KKA$ | 0.1-1 | 0.5 |
| $\alpha^{IFL}$ | - | Spatial heterogeneous coefficient for infiltration capacity | 0.1-5 | 1.5 |
| $\alpha^{EFL}$ | - | Spatial heterogeneous coefficient for exfiltration capacity | 0.1-20 | 0.7 |
| $\alpha^{ETL}$ | - | Spatial heterogeneous coefficient for evapotranspiration capacity | 0.1-5 | 0.7 |
| $DDFg$ | mm℃day⁻¹ | Degree day factor glacier | 0-15 | 6.0 |
| $DDFs$ | mm℃day⁻¹ | Degree day factor snow | 0-15 | 4.8 |





Table 4. Annual mean observed discharge and simulated discharge forced by WFD and native/corrected

RCMs datasets at the Bahadurabad station.

| Discharge | Calibration period | | | | Validation period | | | |
|---|---|---|---|---|---|---|---|---|
| $10^4$m$^3$/s | native | QM | JBCp | JBCt | native | QM | JBCp | JBCt |
| obs | 2.23 | | | | 2.29 | | | |
| WFD | 2.08 | | | | 2.09 | | | |
| RCM1 | 3.12 | 2.01 | 2.07 | 1.97 | 3.23 | 2.11 | 2.16 | 2.07 |
| RCM2 | 2.73 | 2.03 | 2.05 | 2.00 | 2.85 | 2.12 | 2.15 | 2.09 |
| RCM3 | 1.80 | 2.34 | 2.31 | 2.55 | 1.84 | 2.37 | 2.33 | 2.61 |
| RCM4 | 1.88 | 2.24 | 2.25 | 2.41 | 1.92 | 2.27 | 2.28 | 2.45 |
| RCM5 | 2.02 | 1.87 | 1.89 | 1.90 | 2.24 | 2.08 | 2.10 | 2.13 |








Table 5. Nash-Sutcliffe efficiency coefficient (NSE) of streamflow simulation forced by WFD and native/corrected RCMs datasets at daily and

monthly time scales (denoted as day and mon in the table).

| NSE | RCM1 | | | | RCM2 | | | | RCM3 | | | | RCM4 | | | | RCM5 | | | |
|---|---|---|---|---|---|---|---|---|---|---|---|---|---|---|---|---|---|---|---|---|
| | calibration | | validation | | calibration | | validation | | calibration | | validation | | calibration | | validation | | calibration | | validation | |
| | day | mon | day | mon | day | mon | day | mon | day | mon | day | mon | day | mon | day | mon | day | mon | day | mon |
| WFD | 0.84 | 0.92 | 0.78 | 0.84 | | | | | | | | | | | | | | | | |
| RCM | -0.1 | 0.10 | -0.0 | 0.17 | 0.46 | 0.61 | 0.39 | 0.51 | 0.52 | 0.64 | 0.40 | 0.53 | 0.56 | 0.70 | 0.56 | 0.67 | 0.56 | 0.69 | 0.54 | 0.70 |
| RCM_QM | 0.53 | 0.66 | 0.56 | 0.66 | 0.51 | 0.63 | 0.47 | 0.57 | 0.57 | 0.69 | 0.44 | 0.58 | 0.56 | 0.72 | 0.58 | 0.70 | 0.41 | 0.51 | 0.51 | 0.63 |
| RCM_JBCp | 0.56 | 0.69 | 0.58 | 0.69 | 0.53 | 0.66 | 0.49 | 0.60 | 0.58 | 0.71 | 0.46 | 0.60 | 0.57 | 0.72 | 0.59 | 0.70 | 0.42 | 0.52 | 0.51 | 0.63 |
| RCM_JBCt | 0.44 | 0.56 | 0.50 | 0.60 | 0.39 | 0.50 | 0.35 | 0.43 | 0.59 | 0.72 | 0.51 | 0.65 | 0.60 | 0.76 | 0.64 | 0.75 | 0.49 | 0.59 | 0.56 | 0.69 |






Table 6. Evaluation merits of streamflow simulations for individual RCM and BMA scenarios.

| Scenarios | | Calibration | | | Validation | | |
|---|---|---|---|---|---|---|---|
| | | NSE | RE | RMSE | NSE | RE | RMSE |
| | | | (%) | (m³/s) | | (%) | (m³/s) |
| QM | RCM1 | 0.53 | 9.9 | 12070.7 | 0.56 | 7.8 | 12519.3 |
| | RCM2 | 0.51 | 9.0 | 12312.7 | 0.47 | 7.1 | 13701.0 |
| | RCM3 | 0.57 | -4.9 | 11573.7 | 0.44 | -3.8 | 14158.6 |
| | RCM4 | 0.56 | -0.5 | 11633.8 | 0.58 | 0.5 | 12174.1 |
| | RCM5 | 0.41 | 16.3 | 13487.3 | 0.51 | 8.9 | 13269.3 |
| JBCp | RCM1 | 0.56 | 7.2 | 11703.5 | 0.58 | 5.4 | 12244.0 |
| | RCM2 | 0.53 | 8.1 | 12061.4 | 0.49 | 6.0 | 13424.4 |
| | RCM3 | 0.58 | -3.4 | 11369.7 | 0.46 | -1.9 | 13898.5 |
| | RCM4 | 0.57 | -0.9 | 11568.2 | 0.59 | 0.3 | 12134.9 |
| | RCM5 | 0.42 | 15.4 | 13427.7 | 0.50 | 8.1 | 13264.3 |
| JBCt | RCM1 | 0.44 | 11.9 | 13111.4 | 0.50 | 9.4 | 13374.6 |
| | RCM2 | 0.39 | 10.5 | 13732.9 | 0.35 | 8.5 | 15243.1 |
| | RCM3 | 0.59 | -15.0 | 11204.6 | 0.51 | -14.1 | 13165.9 |
| | RCM4 | 0.60 | -7.9 | 11161.9 | 0.64 | -7.4 | 11347.8 |
| | RCM5 | 0.49 | 15.0 | 12613.0 | 0.62 | 6.9 | 12564.8 |
| BMA | | 0.64 | 6.9 | 10524.2 | 0.61 | 4.8 | 11745.9 |





Table 7. Summary of existing studies on projected streamflow under climate change in the YBR Basin.

| Hydrological model | Study Area, Calibration Hydrological Station | GCMs/RCMs | Scenarios | Bias Correction | Bayesian Model Averaging | Streamflow Change Results | Reference |
|---|---|---|---|---|---|---|---|
| Snowmelt Runoff Model | upper YBR Basin, no calibration station | GCMs (CCMA-CGCM3, GFDL-CM2,MPIM-ECHAM5,NIES-MIROC3, UKMO-HADGEM1) | Obs (2000-2007) A1B (2046-2065) | No | No | -19.6% | Immerzeel et al. (2010) |
| Spatial Processes in Hydrology (SPHY) model | upper YBR Basin, no calibration station | GCMs (RCP4.5:GISS-E2-R, IPSL-CM5A-LR, CCSM4, CanESM2; RCP8.5: GFDL-ESM2G, IPSL-CM5A-LR, CSIRO-Mk3-6-0, CanESM2) | Obs (1998-2007) RCP4.5 (2041-2050) RCP8.5 (2041-2050) | No | No | 4.5%(RCP4.5) 5.2%(RCP8.5) | Lutz et al. (2014) |
| H08 Hydrological model | YBR Basin, Bahadurabad | GCMs (MRI-AGCM3.2S, MIROC5, MIROC-ESM, MRI-CGCM3, HadGEM2-ES) | Obs (1980-2001) Near-future (2015-2039) Far-future (2075-2099) | Yes | No | 6.7%(near-future) 16.2%(far-future) RCP8.5 | Masood et al. (2015) |
| Tsinghua Representative Elementary Watershed (THREW) model | YBR Basin, Bahadurabad | RCMs(HadGEM3-RA, RegCM, SNU-MM5, SNU-WRF, YSU-RSM) | Obs (1980-2001) RCP4.5 (2006-2035) RCP8.5 (2006-2035) | Yes | Yes | 6.8%(RCP4.5) 12.9%(RCP8.5) | This study |




Table 8. Means of precipitation / temperature / runoff in the future period (2020-2035) and their relative changes compared to the historical period
(1980-2001) under different scenarios in the YRB.

|  | P | $R_P$ | T | $R_T$ | R | $R_R$ | $r_R$ | $r_G$ | $r_S$ |
|---|---|---|---|---|---|---|---|---|---|
|  | (mm/a) | (%) | (℃) | (℃) | (mm/a) | (%) |  |  |  |
| His-B | 1425.3 | - | 8.7 | - | 1298.4 | - | 87.0% | 3.2% | 97% |
| fs4.5-B | 1529.8 | 7.3% | 9.8 | 1.1 | 1386.7 | 6.8% | 86.5% | 3.3% | 10.2% |
| fs8.5-B | 1608.0 | 12.8% | 10.0 | 1.3 | 1466.4 | 12.9% | 86.9% | 3.2% | 10.0% |
| His-O | 668.9 | - | 1.0 | - | 611.6 | - | 68.9% | 9.0% | 22.1% |
| fs4.5-O | 639.9 | -4.4% | 2.2 | 1.3 | 609.3 | -0.4% | 64.4% | 9.9% | 25.7% |
| fs8.5-O | 748.3 | 11.9% | 2.6 | 1.6 | 691.9 | 13.1% | 67.4% | 9.0% | 23.6% |
| His-N | 631.6 | - | -0.1 | - | 485.8 | - | 74.4% | 5.3% | 20.3% |
| fs4.5-N | 595.8 | -5.7% | 1.2 | 1.3 | 465.8 | -4.1% | 69.3% | 6.1% | 24.6% |
| fs8.5-N | 712.0 | 12.7% | 1.6 | 1.7 | 582.5 | 19.9% | 74.8% | 5.0% | 20.3% |

Note: P denotes precipitation, T denotes temperature, R denotes runoff; $R_P$, $R_T$, $R_R$ denote relative changes of P, T and R compared to
the historical period, respectively; $r_R$, $r_G$, $r_S$ denotes the ratio of rainfall, glacier melting, and snow melting induced runoff in the total
runoff, respectively; -B denotes Bahadurabad, -O denotes he upper YBR basin outlet, and -N denotes Nuxia.



# List of Figures





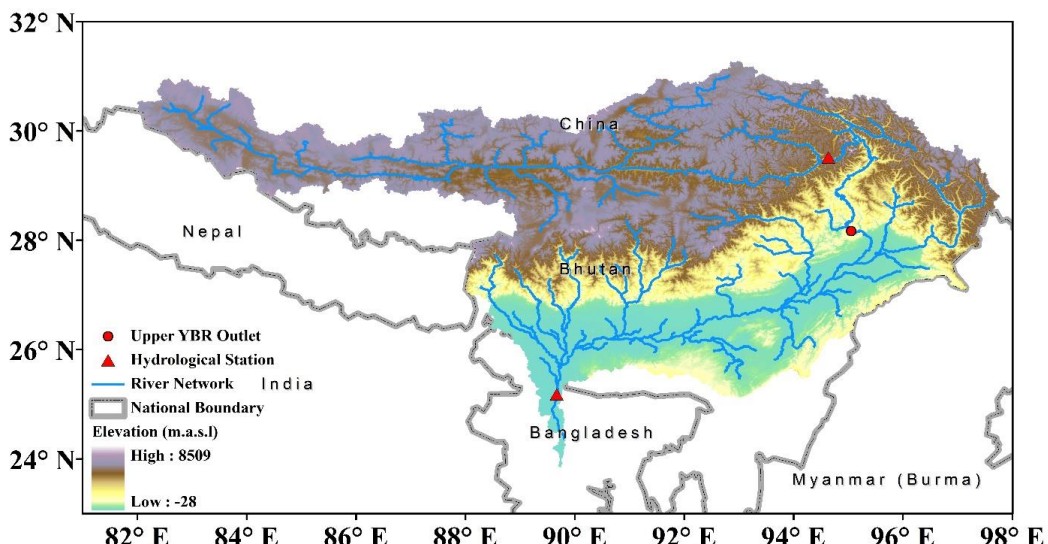


Figure 1. Study area, river network and location of hydrological stations (Nuxia in the upstream basin,

Bahadurabad in the downstream basin).





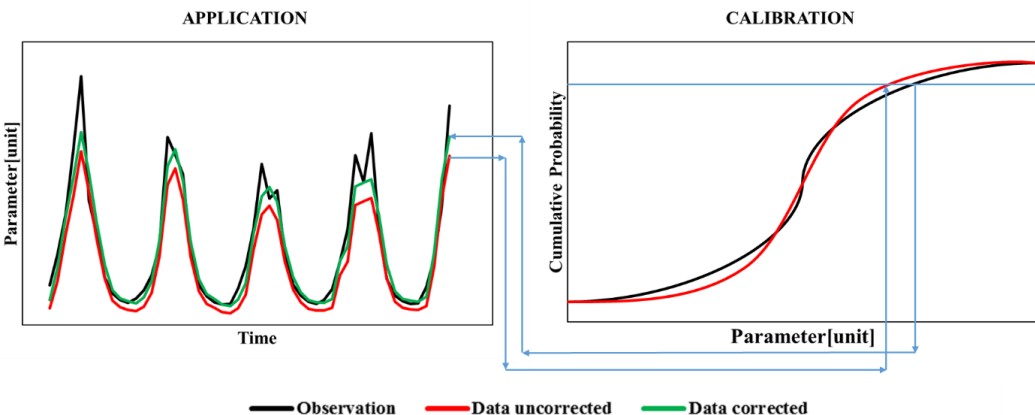


Figure 2. Schematic illustration of quantile mapping bias correction method applied in the paper (Wlicke

et al., 2013).







Figure 3. Seasonal cycles of precipitation from WFD and native/corrected RCMs during the historical
period (1980-2001). (a) for RCM1, (b) for RCM2, (c) for RCM3, (d) for RCM4, (e) for RCM5.





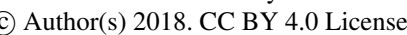

Figure 4. Seasonal cycles of temperature from WFD and native/corrected RCMs during the historical

period (1980-2001). (a) for RCM1, (b) for RCM2, (c) for RCM3, (d) for RCM4, (e) for RCM5.


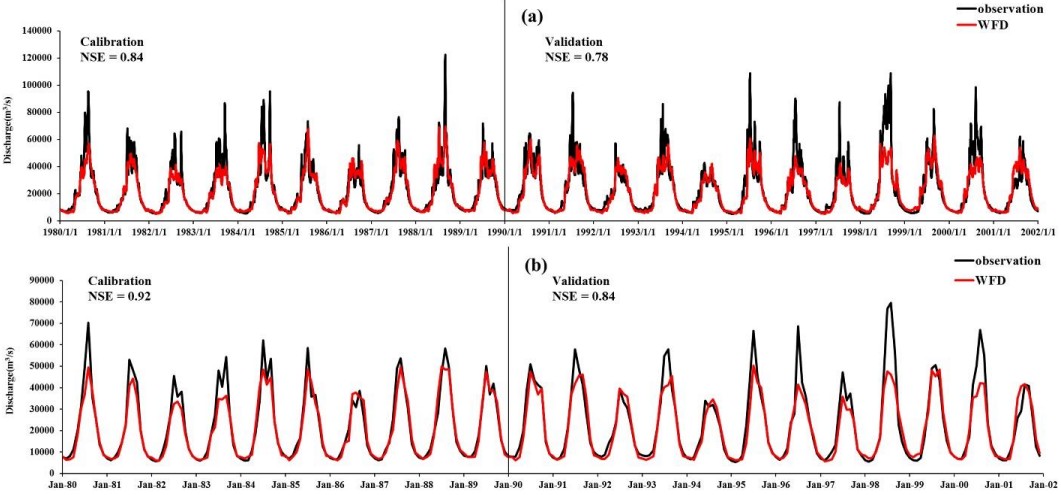


Figure 5. The simulated (red line) and observed (black line) discharge at Bahadurabad at the (a) daily

scale, (b) monthly scale.






Figure 6. Seasonal cycles of observed streamflow and simulated streamflow forced by WFD and
native/corrected RCMs during the calibration period (left column) and validation period (right column) at
Bahadurabad.



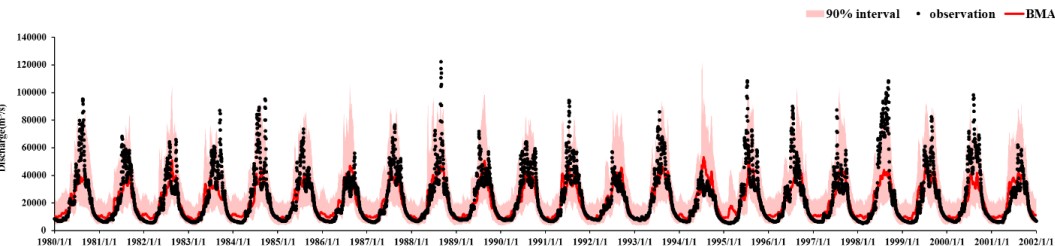


Figure 7. The mean values and 90% uncertainty interval of streamflow simulated by the BMA method

during the historical period.




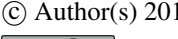

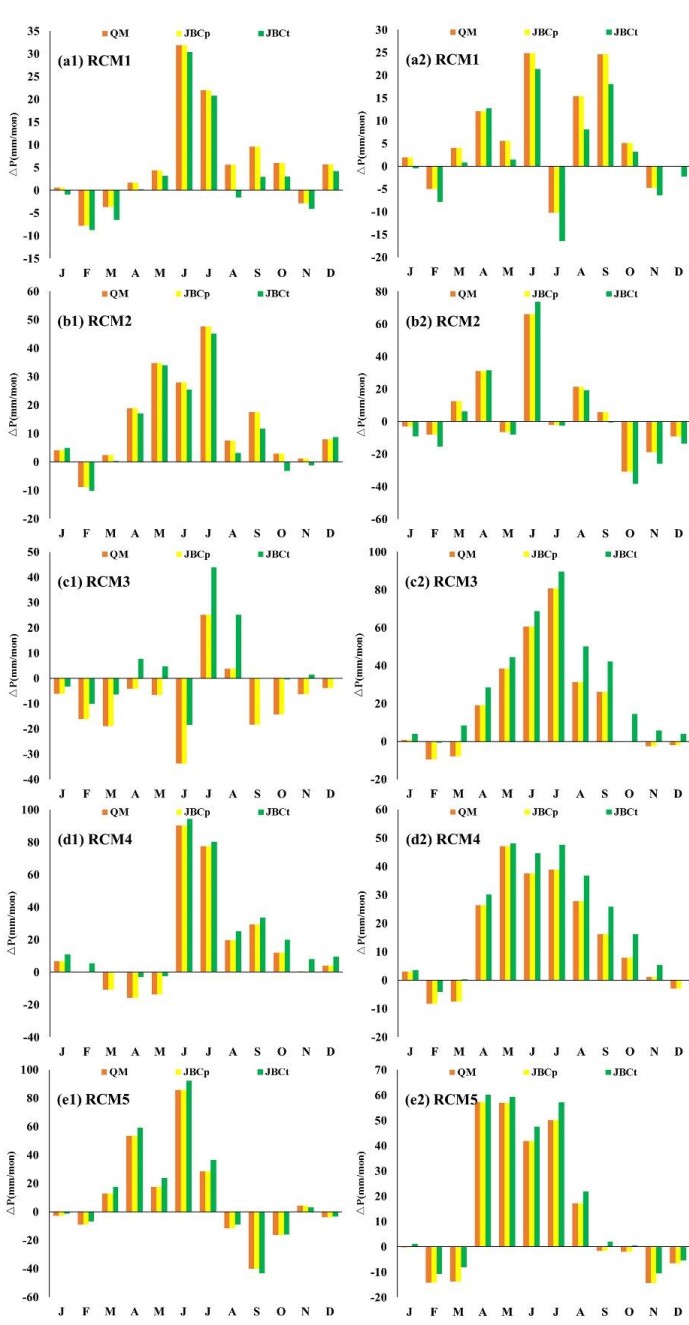


Figure 8. Change of basin-wide precipitation in the future period projected by corrected RCMs under

RCP4.5 (left column) and RCP8.5 (right column) scenarios compared to the historical period.




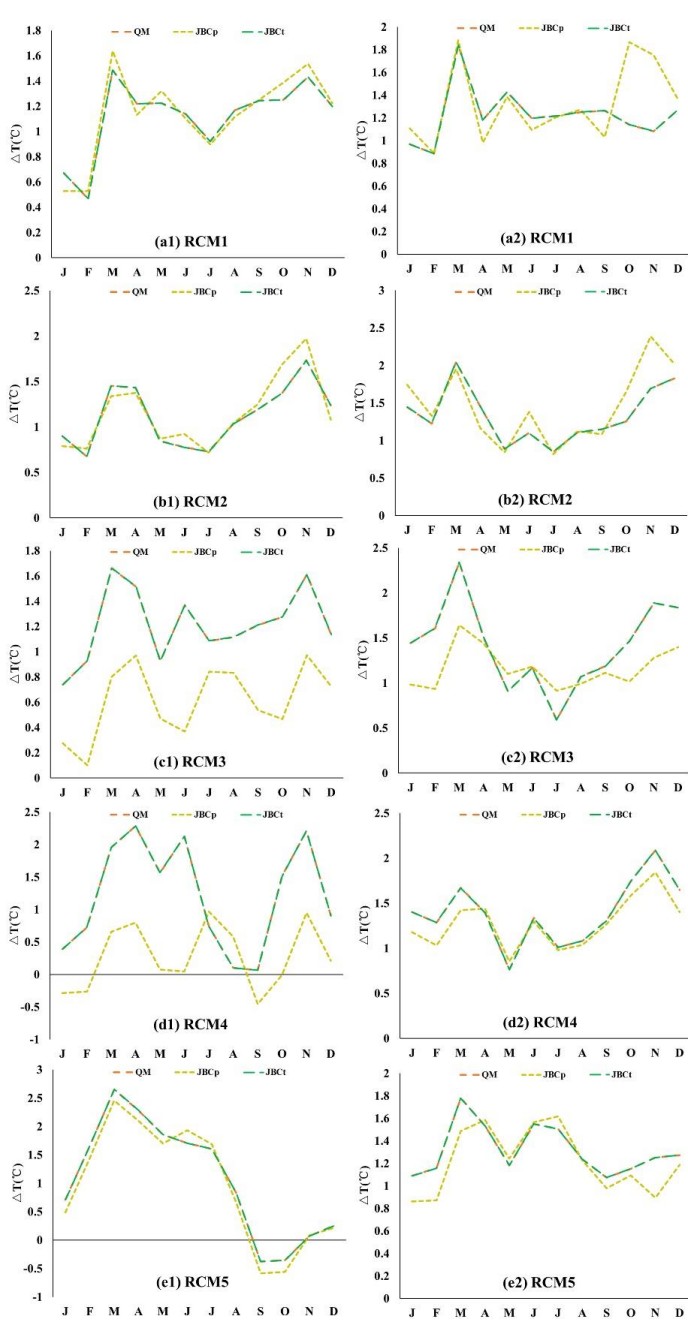


Figure 9. Change of basin-wide temperature in the future period projected by corrected RCMs under

RCP4.5 (left column) and RCP8.5 (right column) scenarios compared to the historical period.






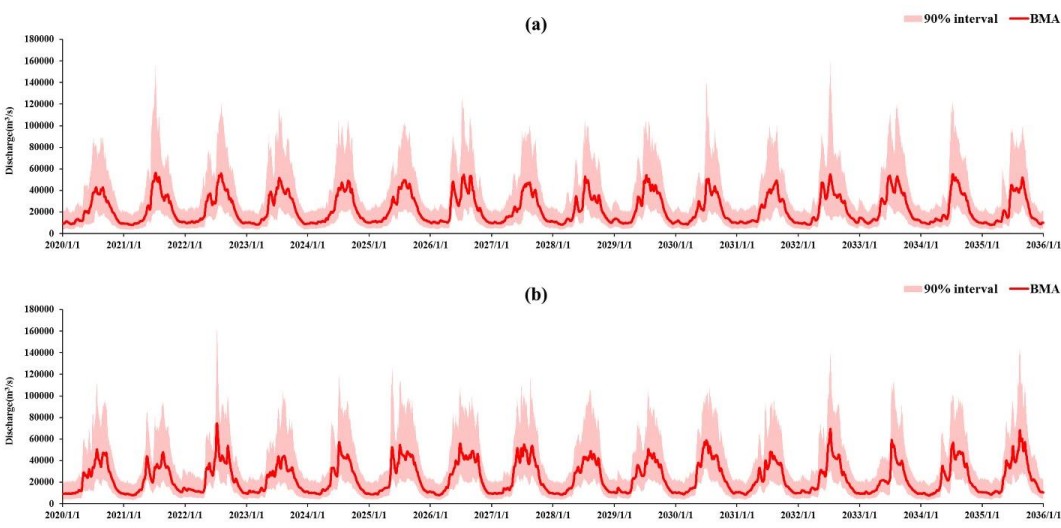


Figure 10. The mean values and 90% uncertainty interval of streamflow simulated by the BMA method

during the future period under (a) RCP4.5, (b) RCP8.5 scenarios at Bahadurabad.








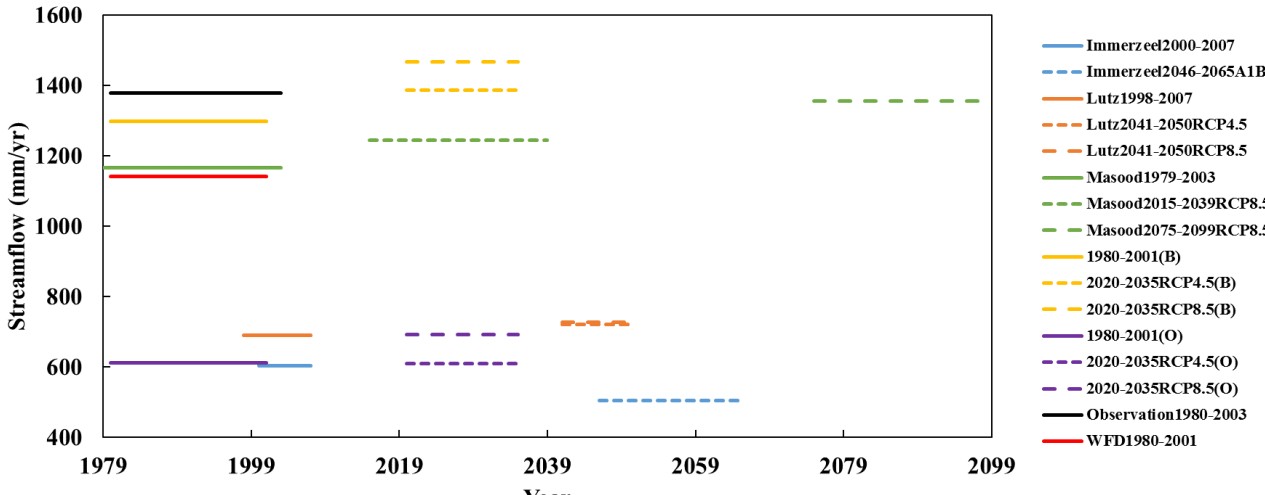

Figure 11. Streamflow projections from the existing studies during different periods at different locations (B denotes Bahadurabad in the
downstream, O denotes the upper YBR basin outlet, see Figure 1 for the location).