# Peer review of "Projected Climate Change Impacts on Future Streamflow of the Yarlung Tsangpo-Brahmaputra River Ran Xu1, Hongchang Hu1, Fuqiang Tian1\*, Chao Li2, Mohd Yawar Ali Khan1"

_Hydrology and Earth System Sciences, 2018_

## Referee Comment (RC1) · Anonymous Referee #1 · 12 Jun 2018

General Comments The work displays a clear and replicable framework to assess variations in streamflows under the potential effect of climate changes. In this regard, the uncertainties associated to different elements of simulation chains (climate models, bias correction approaches) are properly taken into account. Finally, the findings are compared with those retrievable on the same area in an effective way. Nevertheless, some details should be clarified as reported in specific comments to permit the publication. In my opinion, they may contribute to define in clear way all the benefits and constraints associated to adopted framework and in which ways it could be improved.

Specific Comments: L42: at the moment, several models provide data up to 12 km

(e.g. EURO-CORDEX); please add details about them

L49: please prefer "weather" to "climatic"

L58: please mention also several works in which constraints and weaknesses of Bias Correction approaches are clearly stressed (e.g. Ehret et al., 2012)

Ehret, U., Zehe, E., Wulfmeyer, V., Warrach-Sagi, K., and Liebert, J.: HESS Opinions "Should we apply bias correction to global and regional climate model data?", Hydrol. Earth Syst. Sci., 16, 3391-3404, https://doi.org/10.5194/hess-16-3391-2012, 2012.

Figure 1: please another colour for the borders

L108: I suggest you introducing in this context Figure 3 where observed monthly data are displayed

L111: please report the resolution of the dataset also in the text and not only in Table 1

L115: please report brief notes about how calculating PET

L119: do you consider snow cover as an input data for the modelling?

L120: are you sure that land use is not strongly changed moving from reference period (1980-2001) to the period used for inputs?

L135: please refer to RCP as "concentration" and not "emission" scenarios

L137: why do you use 16 years for the future? According WMO indications, 30 years should be the standard to properly taken into account interannual variability. Moreover, on the reference current period 20 years are used

L141-145: the climate simulation chains are the only available under CORDEX initiative on the area of interest? What is the domain?

Equation (1): how PET0 is computed?

L163: brief details about the approach should be included for Hock (2003)

L176: please add references about the method for Bias Correction? Are they able to maintain the climate signal as provided by raw climate simulations?

L188: in which way, are they weighted?

Table 3:the soil in unsaturated and saturated zones is the same or does it have only the same saturated hydraulic conductivity? Why soils characterized by very different porosities have the same hydraulic conductivities? Why do you assume that water table has fixed depth? Which literature method do you use for assessing infiltration and exfiltration capacities?

Figure 3: according your view, why the findings after bias correction are not so good in several months for RCM5? JBCt approach returns in some cases poor performances (worse than the raw climate modelling chains). Given the relevance of precipitation in hydrological modelling, do you consider that it could affect in relevant way the analysis ?

Figure 5 (a-b): in terms of peaks, the analysis is not able to properly reproduce them; in my view, it is due to use of gridded datasets at low resolution; what is your opinion? Are there some weaknesses in parameterizations for hydrological modelling inducing them?

Figure 5 (c-d) are missing

L287: please prefer "weather" to "meteorological"

Figure 6: you should report also results by using WFD as input data for hydrological modelling; indeed, they represent your reference data to understand in which way climate simulation chains can affect the proper reproduction of observed patterns

Figure 8:for precipitation percentage anomaly could be added to have higher information content ; moreover anomalies detected by raw climate simulations should be included; indeed maintaining the signal provided by them represent a key issue when adopting bias correction approaches (the same also for temperature in Figure 9)

Table 6: probably, as in many cases, raw data have performances comparable to BC ones; could they be included in BMA according your view?

L304: it should be stressed the most significant variations among scenarios are expected in the end of the century while you consider medium time horizon

L312: please use "projections" and not "predictions"

L341-342: please add details on these issue (how snow and glacier melting processes can affect the processes

---

## Referee Comment (RC2) · Anonymous Referee #2 · 9 Jul 2018

The authors feed a hydrological model implemented in the Yarlung Tsangpo-Brahmaputra River basin with climate projection provided by the CORDEX project. The CORDEX projections were simulated with regional climate models with grid-spacing of 0.44°, were driven by some global climate projections, and are uncertain. The manuscript aims at reducing the impact of this uncertainty in runoff projections. Uncertainty reduction is tried with the application of bias reduction methods and by combining the different bias corrected projections using the Bayesian model averaging (BMA) method. The combination of methods and the brief review of existing studies in the area makes the manuscript potentially interesting, but there are substantial open questions left:

1) Daily simulation data from five regional climate models (RCMs) are used. The information about the driving global data is not given. Which global climate projections/models drive the RCMs? The GCMs are essential because the uncertainty chain is initiated at global scales already. If, for example, all five RCMs were driven by the same global projection, then the uncertainties are substantially underestimated. Additionally, application of BMA at daily basis needs temporal coincidence between simulations and observational reference data. Therefore, at least reanalysis-driven RCM simulations have to be applied.

2) The data used in the three bias correction methods are not seasonally stratified. This lack of stratification is on one hand sub-optimal as the authors found seasonally dependent biases (which is not surprising given the different precipitation generating mechanisms in winter and during the monsoon period). On the other hand, the sample size of precipitation days in non-monsoon seasons is limiting the quality of bias correction methods (Dobler & Ahrens 2008). Figure 6, bottom row, illustrates this bias correction challenge very well: the non-corrected RCM5 provides much better input into the rainfall-runoff model than the with bias correction. Also, the other panels show bias correction difficulties. There is a tendency that corrected input yields a change of sign in bias from non-monsoonal to monsoonal periods. Therefore, the question is if bias correction introduces systematic errors into the precipitation and temperature data and what can justify the application of bias correction in this basin for future projections?

3) I like the idea of an optimal combination of projections. I am skeptical that BMA is the best choice and this should be investigated much more in depth as a selling point of this manuscript. First of all, it would be an add-on to include the output of the driving global models in the multi-model ensemble (enlarge the ensemble and at the same time show the added value of RCMs). However, BMA needs coincidence in time, and thus the weights of global climate projections cannot be estimated. Second, the bias-corrected output still has a bias (see above). Bias hinders the application of BMA. Third, there are much simpler and more robust methods (like equal weighting

or weighting with some simple skill measure which does not need coincidence, e.g., Casanova and Ahrens 2009).

In general, the data and methods applied should be described much more apprehensively and esp. the weighting method also in more depth. Some parts of the text and figures are not easy to follow. For example, Fig. 11 is very confusing: what RCP, what period, what basin, what author? I also suggest doing much more literature research: what is done in other basins and even for the Yarlung Tsangpo-Brahmaputra River there is more literature to be considered.

Casanova, S., B. Ahrens (2009). On the weighting of multi-model ensembles in seasonal and short-range weather forecasting. Mon. Wea. Rev., 137, 3811-3822, DOI: 10.1175/2009MWR2893.1

Dobler, A., B. Ahrens (2008). Precipitation by a regional climate model and bias correction in Europe and South-Asia. Meteorol. Zeitschrift, 17(4), 499-509. DOI: 10.1127/0941-2948/2008/0306

---

## Author Comment (AC1) · 4 Aug 2018

**Responses to Reviewer 1**

*General Comments The work displays a clear and replicable framework to assess variations in streamflows under the potential effect of climate changes. In this regard, the uncertainties associated to different elements of simulation chains (climate models, bias correction approaches) are properly taken into account. Finally, the findings are compared with those retrievable on the same area in an effective way. Nevertheless, some details should be clarified as reported in specific comments to permit the publication. In my opinion, they may contribute to define in clear way all the benefits and constraints associated to adopted framework and in which ways it could be improved.*

**_Response:_** We would like to express our sincere thanks to the anonymous reviewer for the insightful and detailed comments on our submitted manuscript. We have revised the manuscript thoroughly based on these comments, and address them below on a point-by-point basis.

*(1) L42: at the moment, several models provide data up to 12 km (e.g. EURO-CORDEX); please add details about them.*

**_Response:_** Done. Thanks!

Climate simulations from GCMs can be dynamically downscaled with regional climate models (RCMs) to scales as fine as 4 km. Liu et al. (2016) presented a high resolution climate change simulation at 4-km grid using the Weather Research and Forecasting (WRF) model spacing over much of North America. Jacob et al. (2014) established a new high-resolution regional climate change ensemble with a horizontal resolution of 12.5 km for Europe within the World Climate Research Program Coordinated Regional Downscaling Experiment (EURO-CORDEX).

*(2) L49: please prefer "weather" to "climatic".*

**_Response:_** Done. Thanks!

*(3) L58: please mention also several works in which constraints and weaknesses of Bias*

*Correction approaches are clearly stressed (e.g. Ehret et al., 2012).*

**_Response:_** Done. Thanks!

A more comprehensive review of the constraints of individual bias correction can be found in Ehret et al. (2012).

*(4) Figure 1: please another colour for the borders.*

**_Response:_** Done. Thanks!

[Figure]

*(5) L108: I suggest you introducing in this context Figure 3 where observed monthly data are displayed.*

**_Response:_** Done. Thanks!

*(6) L111: please report the resolution of the dataset also in the text and not only in Table 1.*

*Response:* Done. Thanks!

*(7) L115: please report brief notes about how calculating PET.*

*Response:* The Penman–Monteith equation was used for calculating *PET* in this dataset (Weedon et al., 2011). Thanks!

*(8) L119: do you consider snow cover as an input data for the modelling?*

*Response:* Yes. The MODIS snow covered area data were used. Daily snow cover data were obtained by linear interpolation of the 8-day data. Snow melt were simulated by a degree-day model with the degree-day factor. Details about snow cover as an input data for the modelling could be found in He et al. (2015). Thanks!

*(9) L120: are you sure that land use is not strongly changed moving from reference period (1980-2001) to the period used for inputs?*

*Response:* As we all know, land use influenced significantly by human activities and climate change (Cui and Graf, 2009). The YBR is one of the largest rivers originating from the TP in Southwest China at an elevation of about 3100 m above sea level (Goswami, 1985). Li et al. (2012) found that only 1 % of the land cover in the basin changed during 1985-2005. The high altitude and environmental policy of China made this study area little impacted by human activities. Thanks!

*(10) L135: please refer to RCP as "concentration" and not "emission" scenarios.*

*Response:* We changed all the "emission scenarios" to "concentration scenarios" in the revised manuscript. Thanks!

*(11) L137: why do you use 16 years for the future? According WMO indications, 30 years should be the standard to properly taken into account interannual variability. Moreover, on the reference current period 20 years are used.*

*Response:* We agree with the reviewer that a longer data period would be the most desirable. Comparing with 1986-2005, global average surface temperature would increase 0.3-0.7℃ under RCP2.6 during 2016-2035 (Ofipcc, 2013). However, the CORDEX experiment for the East Asia domain contained 5 models, which was shown in Table 1 in the manuscript. The time series of these 5 models were shown in Table I. 2006-2100 of HadGEM3-RA under RCP4.5 and 8.5, 2006-2050 of RegCM and YSU-RSM under RCP4.5 and 8.5, 2006-2049 of SNU-WRF under RCP4.5 and 8.5, 2006-2049 of SNU-MM5 under RCP4.5 while 2006-2035 of SNU-MM5 under RCP8.5. To compare more RCMs and more concentration scenarios (RCP4.5 and 8.5), the longest overlap time of these 5 RCMs under RCP4.5 and 8.5 was chosen, that from now to 2035. What's more, there were also some researches of projected climate change impacts using less than 30 years for the future (Immerzeel et al., 2010; Lutz et al., 2014). Thanks!

Table I The time series of these 5 RCMs.

| Model | HadGEM3-RA | RegCM | SNU-MM5 | SNU-WRF | YSU-RSM |
|---|---|---|---|---|---|
| History | 1950-2005 | 1979-2005 | 1979-2005 | 1979-2005 | 1980-2005 |
| RCP4.5 | 2006-2100 | 2006-2050 | 2006-2049 | 2006-2049 | 2006-2050 |
| RCP8.5 | 2006-2100 | 2006-2050 | 2006-2035 | 2006-2049 | 2006-2050 |

*(12) L141-145: the climate simulation chains are the only available under CORDEX initiative on the area of interest? What is the domain?*

*Response:* The domain of East Asia could be found from http://www.cordex.org/domains/region-7-east-asia/. The model domain includes East Asia, India, the Western Pacific Ocean, and the northern part of Australia. Specifically, the region presented was defined by (a) parameters needed by an RCM using a rotated pole coordinate system and (b) parameters for RCM using other system coordinates (in non-rotated coordinates). Parameters of East Asia were shown in the link above. And East Asia domain of CORDEX covered the whole YBR Basin. Thanks!

*(13) Equation (1): how PET0 is computed?*

*Response: PET0* was the potential evapotranspiration data from 1980 to 2001 of WATCH forcing data. As mentioned in Response (7), the Penman–Monteith equation was used for calculating *PET* in this dataset (Weedon et al., 2011). We also added this information in the revised manuscript. Thanks!

*(14) L163: brief details about the approach should be included for Hock (2003).*

**_Response:_** For the simulation of snow and glacier melting processes which is important for the YBR Basin, we modified the original THREW model by incorporating the temperature-index method introduced in Hock (2003), that related ice or snow melt to air temperature using degree-day factors. Since air temperature was the most easily available data, this model had been the most widely used method of ice and snow melt computations for many purposes, e.g. hydrological modelling.  Thanks!

*(15) L176: please add references about the method for Bias Correction? Are they able to maintain the climate signal as provided by raw climate simulations?*

**_Response:_** The reference was mentioned before in the manuscript: Quantile mapping (QM) with reference observations has been routinely applied to correct biases in RCM simulations (Maraun, 2013). According to the reference, this method might affect trends of data. Changes in future mean were likely to be misrepresented. To increase the signal-to-noise ratio, one often averages neighboring grid boxes. Thanks!

*(16) L188: in which way, are they weighted?*

**_Response:_** According to Hewitson and Crane (1996), Bayesian model averaging (BMA) mean prediction was a weighted average, with their posterior probabilities being the weights, of the individual model's predictions. Thanks!

*(17) Table 3: the soil in unsaturated and saturated zones is the same or does it have only the same saturated hydraulic conductivity? Why soils characterized by very different porosities have the same hydraulic conductivities? Why do you assume that water table has fixed depth? Which literature method do you use for assessing infiltration and exfiltration capacities?*

**_Response:_** In our opinion, it was the result of calibrating, that the soil in unsaturated and saturated zones had the same saturated hydraulic conductivity. Method of Reggiani et al. (1999) was used for assessing infiltration and exfiltration capacities. Similar to Reggiani et al.'s

definition, the saturated zone is delimited by the water table on the top and by a limit depth reaching into the groundwater reservoir or by the presence of an impermeable stratum at the bottom. Thanks!

*(18) Figure 3: according your view, why the findings after bias correction are not so good in several months for RCM5? JBCt approach returns in some cases poor performances (worse than the raw climate modelling chains). Given the relevance of precipitation in hydrological modelling, do you consider that it could affect in relevant way the analysis?*

**_Response_**_:_ The seasonal cycle of precipitation of RCM5 was not so obvious comparing with other RCMs, i.e. total precipitation of each month showed little differences. What's more, dry seasons and rainy seasons did not show significantly difference. As for JBCt approach, which corrects temperature first and then precipitation, would returned in some cases poor performances (worse than the raw climate modelling chains), e.g. RCM3 and RCM4 in rainy season. In line with previous studies (Li et al., 2014; Piani et al., 2010), we also see that bias correction may increase bias in some cases, particularly for wet season precipitation. This poor performances were also shown in hydrological modelling, Figure 6 (c1) – (d2) in the manuscript. Discharges of JBCt during these period were large than the others during the same period. Thanks!

*(19) Figure 5 (a-b): in terms of peaks, the analysis is not able to properly reproduce them; in my view, it is due to use of gridded datasets at low resolution; what is your opinion? Are there some weaknesses in parameterizations for hydrological modelling inducing them?*

**_Response_**_:_ There is a tendency that the THREW model underestimates high peak flows (red vs. black lines in Figure 5). This is partly because of the use of the gridded forcing data which represent averaged values over a low resolution grid box. Such a tendency of underestimation indicates that the reported projections of future streamflow is less useful for assessing climate change impacts on the flood risk in this river basin. Thanks!

*(20) Figure 5 (c-d) are missing.*

***Response****:* Done. We removed the sentence "and (c-d) seasonal time scales for both the calibration and validation periods" and Figure 5 (c-d) in the revised manuscript because these two figures didn't contain more information than Figure 5 (a-b). Thanks!

*(21) L287: please prefer "weather" to "meteorological".*

***Response****:* Done. We changed all the "meteorological variables" to "weather variables" in the revised manuscript. Thanks!

*(22) Figure 6: you should report also results by using WFD as input data for hydrological modelling; indeed, they represent your reference data to understand in which way climate simulation chains can affect the proper reproduction of observed patterns.*

***Response****:* We have completed Figure 6 as the Reviewer suggested in the revised manuscript. Thanks!

*(23) Figure 8:for precipitation percentage anomaly could be added to have higher information content ; moreover anomalies detected by raw climate simulations should be included; indeed maintaining the signal provided by them represent a key issue when adopting bias correction approaches (the same also for temperature in Figure 9).*

***Response****:* We have completed Figure 8-9 as the Reviewer suggested in the revised manuscript. Thanks!

*(24) Table 6: probably, as in many cases, raw data have performances comparable to BC ones; could they be included in BMA according your view?*

***Response****:* We found that precipitation was larger than observation before bias correction. While hydrological model tended to underestimate runoff when precipitation was large. So raw data might have good performances. But this was the effect of bias cancellation. Therefore raw data could not be included in BMA according our view. Thanks!

*(25) L304: it should be stressed the most significant variations among scenarios are expected in the end of the century while you consider medium time horizon.*

**_Response:_** Done. Thanks!

*(26) L312: please use "projections" and not "predictions".*

**_Response:_** Done. Thanks!

*(27) L341-342: please add details on these issue (how snow and glacier melting processes can affect the processes.*

**_Response:_** The proportion of glaciers covered were 2.7%, 5.2% and 3.5% for Nuxia, the upper YBR outlet, and Bahadurabad location, respectively. What's more, the study area was perennial snow area. Glacier melting and snow melting were the important component of runoff. Therefore, glacier and snow melting plays an important and essential part of streamflow. Thanks!

*References:*

Cui, X. and Graf, H.-F.: Recent land cover changes on the Tibetan Plateau: a review, Clim. Change, 94, 47-61, 2009.

Ehret, U., Zehe, E., Wulfmeyer, V., Warrach-Sagi, K., and Liebert, J.: HESS Opinions "Should we apply bias correction to global and regional climate model data?", Hydrol. Earth Syst. Sci., 16, 3391-3404, 2012.

Goswami, D. C.: Brahmaputra River, Assam, India: Physiography, basin denudation, and channel aggradation, Water Resour. Res., 7, 959-978, 1985.

He, Z. H., Tian, F. Q., Gupta, H. V., Hu, H. C., and Hu, H. P.: Diagnostic calibration of a hydrological model in a mountain area by hydrograph partitioning, Hydrol. Earth Syst. Sci., 19, 1807-1826, 2015.

Hewitson, B. C. and Crane, R. G.: Climate downscaling: techniques and application, Clim. Res., 7, 85-95, 1996.

Hock, R.: Temperature index melt modelling in mountain areas, J. Hydrol., 282, 104-115, 2003.

Immerzeel, W. W., Van Beek, L. P., and Bierkens, M. F.: Climate change will affect the Asian water towers, Science, 328, 1382-1385, 2010.

Jacob, D., Petersen, J., Eggert, B., Alias, A., Christensen, O. B., Bouwer, L. M., Braun, A., Colette, A., Déqué, M., Georgievski, G., Georgopoulou, E., Gobiet, A., Menut, L., Nikulin, G., Haensler, A., Hempelmann, N., Jones, C., Keuler, K., Kovats, S., Kröner, N., Kotlarski, S., Kriegsmann, A., Martin, E., van Meijgaard, E., Moseley, C., Pfeifer, S., Preuschmann, S., Radermacher, C., Radtke, K., Rechid, D., Rounsevell, M., Samuelsson, P., Somot, S., Soussana, J.-F., Teichmann, C., Valentini, R., Vautard, R., Weber, B., and Yiou, P.: EURO-CORDEX: new high-resolution climate change projections for European impact research, Reg. Environ. Change, 14, 563-578, 2014.

Li, C., Sinha, E., Horton, D. E., Diffenbaugh, N. S., and Michalak, A. M.: Joint bias correction of temperature and precipitation in climate model simulations, J. Geophys. Res. Atmos., 119, 153-162, 2014.

Li, F., Xu, Z., Feng, Y., Liu, M., and Liu, W.: Changes of land cover in the Yarlung Tsangpo River basin from 1985 to 2005, Environmental Earth Sciences, 68, 181-188, 2012.

Liu, C., Ikeda, K., Rasmussen, R., Barlage, M., Newman, A. J., Prein, A. F., Chen, F., Chen, L., Clark, M., Dai, A., Dudhia, J., Eidhammer, T., Gochis, D., Gutmann, E., Kurkute, S., Li, Y., Thompson, G., and Yates, D.: Continental-scale convection-permitting modeling of the current and future climate of North America, Clim. Dyn., 49, 71-95, 2016.

Lutz, A. F., Immerzeel, W. W., Shrestha, A. B., and Bierkens, M. F. P.: Consistent increase in High Asia's runoff due to increasing glacier melt and precipitation, Nature Climate Change, 4, 587-592, 2014.

Maraun, D.: Bias Correction, Quantile Mapping, and Downscaling: Revisiting the Inflation Issue, J. Clim., 26, 2137-2143, 2013.

Ofipcc, W. G. I.: Climate Change 2013: The Physical Science Basis, Contribution of Working, 43, 866-871, 2013.

Piani, C., Weedon, G. P., Best, M., Gomes, S. M., Viterbo, P., Hagemann, S., and Haerter, J. O.: Statistical bias correction of global simulated daily precipitation and temperature for the application of hydrological models, J. Hydrol., 395, 199-215, 2010.

Reggiani, P., Hassanizadeh, S. M., Sivapalan, M., and Gray, W. G.: A unifying framework for watershed thermodynamics: constitutive relationships, Adv. Water Resour., 23, 15-39, 1999.

Weedon, G. P., Gomes, S., Viterbo, P., Shuttleworth, W. J., Blyth, E., Österle, H., Adam, J. C., Bellouin, N., Boucher, O., and Best, M.: Creation of the WATCH Forcing Data and Its Use to Assess Global and Regional Reference Crop Evaporation over Land during the Twentieth Century, J. Hydrometeorol., 12, 823-848, 2011.

---

## Author Comment (AC2) · 4 Aug 2018

**Responses to Reviewer 2**

*The authors feed a hydrological model implemented in the Yarlung Tsangpo-Brahmaputra River basin with climate projection provided by the CORDEX project. The CORDEX projections were simulated with regional climate models with grid-spacing of 0.44◦, were driven by some global climate projections, and are uncertain. The manuscript aims at reducing the impact of this uncertainty in runoff projections. Uncertainty reduction is tried with the application of bias reduction methods and by combining the different bias corrected projections using the Bayesian model averaging (BMA) method. The combination of methods and the brief review of existing studies in the area makes the manuscript potentially interesting, but there are substantial open questions left:*

**Response:** We would like to express our sincere thanks to the anonymous reviewer for the insightful and detailed comments on our submitted manuscript. We have revised the manuscript thoroughly based on these comments, and address them below on a point-by-point basis.

*(1) Daily simulation data from five regional climate models (RCMs) are used. The information about the driving global data is not given. Which global climate projections/models drive the RCMs? The GCMs are essential because the uncertainty chain is initiated at global scales already. If, for example, all five RCMs were driven by the same global projection, then the uncertainties are substantially underestimated. Additionally, application of BMA at daily basis needs temporal coincidence between simulations and observational reference data. Therefore, at least reanalysis-driven RCM simulations have to be applied.*

**Response:** The Coordinated Regional climate Downscaling Experiment (CORDEX) for East Asia (Domain 7) contained 5 RCMs, which were HadGEM3-RA (denoted by RCM1), RegCM4 (RCM2), SNU-MM5 (RCM3), SNU-WRF (RCM4) and YSU-RSM (RCM5). We used them all in our research. All the RCMs were driven from the historical run of the Atmosphere-Ocean coupled Hadley Center Global Environmental Model version 2 (HadGEM2-AO) GCM simulation of the National Institute of Meteorological Research (NIMR) (Baek et al., 2013). Here we are not focused on the reproduction of the observed streamflow time series, but on the

climatology over a long-term period. RCM simulations driven reanalysis are not required here. Additionally, spatial and temporal coincidence between RCM simulations and WFD observational reference data has been accomplished. Climate model integrations were interpolated to the grid of the WFD using the bilinear interpolation method. Temporal resolution of RCMs and WFD were all at daily scale. Thanks!

*(2) The data used in the three bias correction methods are not seasonally stratified. This lack of stratification is on one hand sub-optimal as the authors found seasonally dependent biases (which is not surprising given the different precipitation generating mechanisms in winter and during the monsoon period). On the other hand, the sample size of precipitation days in non-monsoon seasons is limiting the quality of bias correction methods (Dobler and Ahrens, 2008). Figure 6, bottom row, illustrates this bias correction challenge very well: the non-corrected RCM5 provides much better input into the rainfall-runoff model than the with bias correction. Also, the other panels show bias correction difficulties. There is a tendency that corrected input yields a change of sign in bias from non-monsoonal to monsoonal periods. Therefore, the question is if bias correction introduces systematic errors into the precipitation and temperature data and what can justify the application of bias correction in this basin for future projections?*

**Response:** We cannot agree with this too much because there is clear evidence that native simulations substantially overestimate streamflow in dry seasons, but perform reasonably well for streamflow in wet season. The opposite tendency is found for bias corrected simulations. As a consequence, simulations with and without bias correction seemly have comparable performance. Considering that the THREW model underestimate high flows (Figure 5), the good performance of native simulations in reproducing wet season streamflow is very likely a result of the offsets of two errors. Under this case, one should not say that bias correction does not work. What's more, the value of bias correction for assessing the impacts of climate change on water resources depends to certain extent on the adopted impact models. In our study, the projected streamflow is 1466 mm/a during 2020-2035 under RCP8.5 at Bahadurabad, which is substantially higher than the findings of Masood et al. (2015) at the same location, which is 1244 mm per year during 2015-2039 under RCP8.5. The projected streamflow is 692 mm per year during 2020-2035 under RCP8.5 at the upper YBR outlet. This result is quite close to the

findings of Lutz et al. (2014), which is 727 mm per year during 2041-2050 under RCP8.5. Bias correction is implemented for the whole study period but not separately for each season or month. Our data series is too short to separate for each season or month, which will cause higher level of uncertainty because of insufficiency of sample size. Thanks!

*(3) I like the idea of an optimal combination of projections. I am skeptical that BMA is the best choice and this should be investigated much more in depth as a selling point of this manuscript. First of all, it would be an add-on to include the output of the driving global models in the multi-model ensemble (enlarge the ensemble and at the same time show the added value of RCMs). However, BMA needs coincidence in time, and thus the weights of global climate projections cannot be estimated. Second, the bias-corrected output still has a bias (see above). Bias hinders the application of BMA. Third, there are much simpler and more robust methods (like equal weighting or weighting with some simple skill measure which does not need coincidence, e.g. Casanova and Ahrens (2009)).*

**Response**: It is true that regional climate simulations are not designed to reproduce the time series of the observations but the climatology. We will do this work in the revised manuscript. Thanks!

*(4) In general, the data and methods applied should be described much more apprehensively and esp. the weighting method also in more depth. Some parts of the text and figures are not easy to follow. For example, Fig. 11 is very confusing: what RCP, what period, what basin, what author? I also suggest doing much more literature research: what is done in other basins and even for the Yarlung Tsangpo-Brahmaputra River there is more literature to be considered.*

**Response**: The revised manuscript will give expressions to the Reviewer's suggestions. What's more, we will do more literature, like Su et al. (2016) and Warwade et al. (2017), for the Yarlung Tsangpo-Brahmaputra River Basin. Thanks!

*References:*

Baek, H.-J., Lee, J., Lee, H.-S., Hyun, Y.-K., Cho, C., Kwon, W.-T., Marzin, C., Gan, S.-Y., Kim, M.-J., Choi, D.-H., Lee, J., Lee, J., Boo, K.-O., Kang, H.-S., and Byun, Y.-H.: Climate change in the 21st century simulated by HadGEM2-AO under representative concentration pathways, Asia-Pacific Journal of Atmospheric Sciences, 49, 603-618, 2013.

Casanova, S. and Ahrens, B.: On the Weighting of Multimodel Ensembles in Seasonal and Short-Range Weather Forecasting, Mon. Weather Rev., 137, 3811-3822, 2009.

Dobler, A. and Ahrens, B.: Precipitation by a regional climate model and bias correction in Europe and South Asia, Meteorol. Z., 17, 499-509, 2008.

Lutz, A. F., Immerzeel, W. W., Shrestha, A. B., and Bierkens, M. F. P.: Consistent increase in High Asia's runoff due to increasing glacier melt and precipitation, Nature Climate Change, 4, 587-592, 2014.

Masood, M., Yeh, P. J. F., Hanasaki, N., and Takeuchi, K.: Model study of the impacts of future climate change on the hydrology of Ganges–Brahmaputra–Meghna basin, Hydrol. Earth Syst. Sci., 19, 747-770, 2015.

Su, F., Zhang, L., Ou, T., Chen, D., Yao, T., Tong, K., and Qi, Y.: Hydrological response to future climate changes for the major upstream river basins in the Tibetan Plateau, Global Planet. Change, 136, 82-95, 2016.

Warwade, P., Sharma, N., Pandey, A., and Ahrens, B.: Analysis of Climate Variability in a Part of Brahmaputra River Basin in India, Springer Singapore, 2017.

---

## Referee Comment (RC3) · Anonymous Referee #3 · 9 Aug 2018

The authors have done a great job on this research as there are few research articles on this topic (Brahmaputra River & Climate Change). However, reading the paper thoroughly, I find there are room for improvements:

1) I am not convinced why the authors have chosen 2020-2035 as the climate change period. The impacts of climate change on water resources, that we have been observing worldwide, is very much unpredictable/uncertain in the early stage of 21st century. The precipitation projection within that period is very uncertain ( model to model variation). Although the early stage of 21st century may be of interest from the water management point of view, a separate analysis of later part of the 21st century is required

to fulfill the analysis. My strong recommendation would be to consider 2071-2100 as well.

2) Brahmaputra (or Yarlung Tsangpo) is a perennial river and the hydrograph ( especially at Bangladesh location: Bahadurabad) is very steep during monsoon period resulting a huge variation in wet season flow and dry season flow). The enormous stream-flow during monsoon season causes flood in the lower riparian countries ( like Bangladesh). The authors only considered mean annual stream-flow in their analysis while an analysis of maximum annual flows are essential for the completeness of the study.

---

## Author Comment (AC3) · 14 Aug 2018

**Responses to Reviewer 3**

*The authors have done a great job on this research as there are few research articles on this topic (Brahmaputra River & Climate Change). However, reading the paper thoroughly, I find there are room for improvements:*

**Response:** We would like to express our sincere thanks to the anonymous reviewer for the insightful and detailed comments on our submitted manuscript. We have revised the manuscript thoroughly based on these comments, and address them below on a point-by-point basis.

*(1) I am not convinced why the authors have chosen 2020-2035 as the climate change period. The impacts of climate change on water resources, that we have been observing worldwide, is very much unpredictable/uncertain in the early stage of 21st century. The precipitation projection within that period is very uncertain (model to model variation). Although the early stage of 21st century may be of interest from the water management point of view, a separate analysis of later part of the 21st century is required to fulfill the analysis. My strong recommendation would be to consider 2071-2100 as well.*

**Response:**

We agree with the reviewer that a longer data period would be the most desirable. Comparing with 1986-2005, global average surface temperature would increase 0.3-0.7℃ under RCP2.6 during 2016-2035 (Ofipcc, 2013). However, the CORDEX experiment for the East Asia domain contained 5 models, which was shown in Table 1 in the manuscript. The time series of these 5 models were shown in Table I. 2006-2100 of HadGEM3-RA under RCP4.5 and 8.5, 2006-2050 of RegCM and YSU-RSM under RCP4.5 and 8.5, 2006-2049 of SNU-WRF under RCP4.5 and 8.5, 2006-2049 of SNU-MM5 under RCP4.5 while 2006-2035 of SNU-MM5 under RCP8.5. To compare more RCMs and more concentration scenarios (RCP4.5 and 8.5), the longest overlap time of these 5 RCMs under RCP4.5 and 8.5 was chosen, that from now to 2035. Thanks!

Table I The time series of these 5 RCMs.

| Model | HadGEM3-RA | RegCM | SNU-MM5 | SNU-WRF | YSU-RSM |
|---|---|---|---|---|---|
| **History** | 1950-2005 | 1979-2005 | 1979-2005 | 1979-2005 | 1980-2005 |
| **RCP4.5** | 2006-2100 | 2006-2050 | 2006-2049 | 2006-2049 | 2006-2050 |
| **RCP8.5** | 2006-2100 | 2006-2050 | 2006-2035 | 2006-2049 | 2006-2050 |

*(2) Brahmaputra (or Yarlung Tsangpo) is a perennial river and the hydrograph (especially at Bangladesh location: Bahadurabad) is very steep during monsoon period resulting a huge variation in wet season flow and dry season flow). The enormous stream-flow during monsoon season causes flood in the lower riparian countries (like Bangladesh). The authors only considered mean annual stream-flow in their analysis while an analysis of maximum annual flows are essential for the completeness of the study.*

**_Response_:** The revised manuscript will give expressions to the Reviewer's suggestions about floods. Figure I shows the mean projection (red line) and 90% uncertainty interval of BMA during the historical period at Bahadurabad. The uncertainty was larger than annual water resources. Results showed that floods were significantly underestimated because of no matter hydrological model or driven dataset. Thanks!

[Figure]

Figure I. The mean values and 90% uncertainty interval of streamflow simulated by the BMA method during the historical period.

*References:*

Ofipcc, W. G. I.: Climate Change 2013: The Physical Science Basis, Contribution of Working, 43, 866-871, 2013.